# Impact of Pre-Existing Immunity to Influenza on Live-Attenuated Influenza Vaccine (LAIV) Immunogenicity

**DOI:** 10.3390/vaccines8040683

**Published:** 2020-11-16

**Authors:** Sreeja Roy, Clare M. Williams, Danushka K. Wijesundara, Yoichi Furuya

**Affiliations:** 1Department of Immunology and Microbial Disease, Albany Medical College, Albany, NY 12208, USA; roys3@amc.edu (S.R.); williac10@amc.edu (C.M.W.); 2The School of Chemistry and Molecular Biosciences, The Australian Institute for Bioengineering and Nanotechnology, The University of Queensland, Queensland 4072, Australia; d.wijesundara@uq.edu.au

**Keywords:** live attenuated influenza vaccine, pre-existing immunity, pre-existing antibodies, T cell immunogenicity, mucosal responses

## Abstract

During the previous influenza seasons, between 2010 and 2016, the live attenuated influenza vaccine (LAIV) provided variable efficacy against influenza in the U.S., causing the recommendation against the use of the LAIV. In striking contrast, pre-clinical studies have repeatedly demonstrated superior efficacy of LAIV against mismatched influenza viruses, compared to inactivated influenza vaccines (IIV). This disparity in reported vaccine efficacies between pre-clinical and clinical studies may in part be explained by limitations of the animal models of influenza. In particular, the absence of pre-existing immunity in animal models has recently emerged as a potential explanation for the discrepancies between preclinical findings and human studies. This commentary focuses on the potential impact of pre-existing immunity on LAIV induced immunogenicity with an emphasis on cross-protective immunity.

## 1. Protective Immune Correlates of LAIV Compared to IIV

The live attenuated influenza vaccine (LAIV) can be delivered via the natural route of influenza virus infection and replicates in the nasal cells to induce broad immune responses comprising of both cellular and humoral immunity. Although mucosal vaccination with LAIV generates virus-specific cytotoxic CD8^+^ T lymphocytes (CTL), the systemic hemagglutinin (HA)-specific neutralizing antibodies generated are generally less robust compared to inactivated influenza vaccines (IIVs) [1,2,3]. Instead, some studies have shown that local Immunoglobulin (Ig)A antibodies elicited following LAIV vaccination contribute towards suppression of influenza associated morbidity [4,5,6] and protect against heterologous influenza viral strains [7]. In contrast, parenteral administration of IIV elicits excellent systemic IgG and IgM neutralizing antibodies that primarily target the influenza virus HA and neuraminidase (NA) glycoproteins, but promote lower concentrations of neutralizing IgA antibodies in the lung mucosa [8,9,10,11,12]. Since LAIV is a whole virus vaccine, it presents a broader repertoire of antigens, including internal proteins that are highly conserved among influenza A viruses. In contrast, IIV primarily consists of two viral surface antigens: HA and NA, which are highly variable. Consequently, LAIV can elicit a broader immune response and theoretically, superior cross-protective immunity compared to IIV [reviewed in [13]. Indeed, pre-clinical studies in mice have shown that the seasonal trivalent LAIV provides significant protection against influenza, specifically influenza A viruses [14,15] as well as superior T cell-mediated cross-protective efficacy against heterologous H1N1 and H3N2 infections compared to trivalent IIV [16]. Furthermore, monovalent LAIV against H3N2 in pigs afforded antibody-mediated complete protection against antigenically distinct influenza A H3N2 viruses, while only a partial protection was observed by IIV [17].

Compared to IIV, LAIV can better induce CD4^+^, CD8^+^ as well as γδ T cell responses [18,19,20]. Importantly, CD4^+^ helper T cells provide help to B cells and CD8^+^ T cells to promote antiviral adaptive immunity (reviewed in [21]). LAIV can also activate cytotoxic CD8^+^ T cells, which can recognize epitopes from internal influenza proteins, such as nucleoprotein, acid polymerase and matrix protein, that are highly conserved, and therefore contribute towards protection against mismatched, heterologous influenza virus strains [22,23,24]. In addition, cytotoxic CD4^+^ T cells can also respond to core internal proteins. This response has been correlated with enhanced heterotypic immunity and can be induced by LAIV vaccination [13,25]. In contrast, IIV’s capacity to mount potent CD8^+^ T cell responses is less certain due to conflicting reports [18,26,27,28], but IIV can afford enhanced CD4^+^ T cell responses, specifically follicular T helper cells that are important for germinal center B cell differentiation [29,30]. In terms of B cell immunogenicity, LAIV has been shown to induce cross-reactive plasmablasts and antibodies, which can target the conserved epitopes on HA including the stalk region [31,32]. It is generally believed that the antibodies against conserved regions, the stem, and the receptor binding site of HA are broadly cross-reactive among HA subtypes and therefore may play a role in cross-protection [33,34,35,36]. Furthermore, LAIV has been shown to elicit non-neutralizing antibodies against H7N9 strain, which may contribute to cross-protection potentially via antibody-dependent cellular cytotoxicity (ADCC) mediated by fragment crystallizable region (Fc)–Fc receptor (FcR) interactions [36]. In the context of IIV, although it has failed to generate cross-protective efficacy against the 2009 pandemic H1N1 influenza [37,38,39], a study has demonstrated that partial cross-protection mediated by non-neutralizing antibodies and CD8^+^ T cells can be achieved by sequential vaccination with antigenically distinct IIV in animal models [40]. Furthermore, IIV can elicit non-neutralizing antibodies against the highly conserved HA stalk region [31]. Thus, although HA inhibition (HI) antibody titers are a gold standard for correlation of IIV efficacy, especially during pre-clinical LAIV development, measuring cytotoxic T cell responses and mucosal IgA antibodies in addition to HI titers may more accurately predict LAIV efficacy in humans. 

## 2. LAIV Efficacy: Comparing Pre-Clinical and Clinical Studies

Commercially available LAIV and experimental pandemic-LAIVs (pLAIVs) have consistently demonstrated significant protection in animal models such as mice and ferrets (the most attractive small animal models that closely recapitulate human infection) against influenza virus strains of pandemic potential: H1N1, H5N1, H5N2, H7N3, H2N2 and H7N9 [41,42,43,44,45,46,47,48]. These studies have reported significant control of viral replication in the respiratory tract, robust serum HI antibodies, as well as protection against homologous (against vaccine strains) and heterologous strains including mismatched/drifted influenza virus strains [17]. 

Consistent with pre-clinical data, clinical studies with a monovalent 2009 A/H1N1 LAIV demonstrated significant protective efficacy against the 2009 pandemic H1N1 influenza in children [49,50,51]. Furthermore, the trivalent LAIV also showed moderate [52,53,54] to high efficacy against influenza in children [18,55,56,57]. In the latter studies, the trivalent LAIV elicited significant protective efficacy and immunogenicity against all three vaccine components: H1N1, H3N2 and influenza B virus. It was also demonstrated in a side-by-side comparison that trivalent LAIV is superior to IIV against antigenically similar [18,55] and antigenically drifted influenza strains in some flu seasons [56,57]. However, post licensure of quadrivalent LAIV in the 2013–2016 seasons reported that LAIV efficacies varied between 19–50% in the U.S., along with reduced effectiveness against influenza A viruses [58,59,60,61,62,63,64,65,66]. In addition, reported LAIV efficacies have shown considerable variation in countries other than the U.S. For example, in the same season, a single dose of trivalent LAIV elicited significant protective efficacy against influenza A viruses in Bangladesh, whilst showing no protection against influenza A and B viruses in Senegal [67,68]. Furthermore, compared to the U.S., different LAIV efficacy percentages were reported in countries such as Canada, UK, and Finland [63,69,70,71]. Highly variable LAIV efficacy in the recent influenza seasons and discrepancies between pre-clinical and clinical trials of LAIV raise concerns about the predictive value of animal models used for influenza vaccine research. Towards understanding the underlying reasons behind discrepant reports of LAIV efficacies, pre-existing immunity to influenza has emerged as a potential factor influencing LAIV efficacy [72,73,74,75,76,77,78,79,80]. Concordantly, in our recent study, pre-vaccination with IIV was shown to reduce LAIV (both vaccines expressing same antigens) efficacy against heterologous H1N1 virus [81]. Since cross-reactive antibodies were not detected, we speculate a crucial role of pre-existing antibodies in suppression of LAIV-induced T cell responses that have high cross-reactive potential. Hence, in this commentary, we focused on the role of pre-existing immunity in preventing initial LAIV replication and its subsequent impact on cross-protective immunity.

## 3. Impact of Pre-Existing Antibodies on T cell Immunogenicity of LAIV 

One of the key factors that determine the immunogenicity of LAIV is the replicative fitness of the temperature sensitive, live attenuated virus in the upper respiratory tract [82,83,84]. Successful replication of LAIV in nasal epithelia appears to be a prerequisite for facilitating antigen presentation by the major histocompatibility complex-I (MHC-I) to activate CD8^+^ T cells for effective viral clearance [3,18,85]. The thermal instability of H1N1 LAIV strain (A/California/07/2009), causing its defective replication in human nasal cells [86,87], has been associated with reduced efficacy of LAIV against H1N1 virus [88]. Indeed, modification to a more stable, replication competent version of LAIV in subsequent influenza seasons (2017–2018) improved its efficacy [89]. Despite the recent recognition of the importance of LAIV replication in LAIV immunogenicity, little is known about the negative impact of pre-existing immunity on the replicative fitness of LAIV. A clinical trial in Bangladesh correlated higher pre-existing baseline antibodies derived from natural influenza A/H3N2 and B infections with low viral shedding/replication of LAIV [73]. In addition, pre-existing antibodies derived from prior vaccination have been associated with reduced immunogenicity of LAIV [90,91,92,93]. Furthermore, LAIV was more effective in protecting against the incidence of influenza-like illness among U.S. military personnel who had lower vaccination rates (potentially due to low baseline antibody levels) compared to those immunized regularly with either IIV or LAIV [94]. Similarly, LAIV was also more efficacious than IIV in preventing influenza-associated illness in children, presumably due to low levels of pre-existing immunity in children compared to adults [18,56]. It remains to be elucidated how pre-existing immunity can impact LAIV-induced immune responses. Our recent studies demonstrated that prior vaccination with IIV reduces efficacy of LAIV against heterologous influenza [81]. As described above, IIV elicits robust IgG antibodies, but limited IgA and cytotoxic T cell responses, suggesting that the presence of IIV-specific IgG negatively influences LAIV efficacy. Pre-existing antibodies have also been shown to reduce the efficacy of a variety of other live attenuated vaccines against other pathogens [95]. Thus, taken together, we hypothesize that pre-existing antibodies derived from IIV vaccination and/or seasonal influenza infections neutralize LAIV viruses and diminish the establishment of mucosal immunity at the vaccination site. 

Our recent study also indicated that mice vaccinated with IIV followed by LAIV (expressing the same vaccine antigens) are protected against homologous H1N1 virus challenge [81]. Analysis of serum samples showed induction of strain-specific HI antibodies by IIV, LAIV as well as IIV/LAIV. This indicated that neutralizing antibodies confer strain-specific immunity and thus explains why strain-specific immunity remained unaltered in mice vaccinated with IIV followed by LAIV. However, in this study, prior vaccination of mice with IIV inhibited cross-protective efficacy of LAIV against heterologous H1N1 infection. Since cross-reactive HI antibodies were not associated with LAIV-mediated cross-protection, suppression of cross-reactive T cells by pre-existing serum antibodies is the likely underlying mechanism, although this has not been formally tested in our study. Others have reported that prior vaccination with IIV is associated with suppressed T cell responses in animal models. Bodewas et al. showed that HI antibodies derived from IIV vaccination were associated with prevention of cross-reactive CD8^+^ T cells induced by homologous influenza infection in mice and ferrets [96,97]. The same authors also showed that prior IIV vaccination of mice prevented viral replication and induction of interferon (IFN)-γ^+^ CD62L^hi^ CD127^+^ memory CD8^+^ T cells following heterologous infection, crucial for cross-protection against antigenically drifted and non-vaccine strains [98]. Similarly, in humans, pre-existing antibodies derived from prior vaccination with IIV have been correlated with low levels of heterosubtypic immunity mediated by virus-specific T cells [99]. Additionally, in the context of other vaccines, pre-existing anti-vector immunity has been associated with reduced T cell responses upon vaccination with experimental viral vector vaccines. For example, frequent exposure to various human adenovirus subtypes leads to pre-existing antibodies against adenoviral vectors commonly used for vaccine delivery [100]. This minimizes the effectiveness of these vectors in prime–boost vaccination settings especially if the desired outcome is to augment T cell responses following the boost [101,102]. Studies using human viruses as vectors for vaccine delivery in a homologous prime–boost regimen have failed to enhance IFN-γ expressing T cells specific to vaccine antigens [103,104,105]. These studies suggest a crucial role of pre-existing anti-vector immunity, most likely neutralizing antibodies, in the suppression of antigen specific cell-mediated immunity. Thus, it is imperative to understand immune mechanisms underlying how pre-existing antibodies impact LAIV-induced T cell responses so that better vaccination strategies can be developed. This is particularly true now given the ongoing COVID-19 pandemic. One of the COVID-19 vaccines under development in Hong Kong uses live attenuated influenza virus expressing SARS-CoV-2 antigens [106]. We speculate that anti-vector immunity to influenza may also be detrimental to the immunogenicity of this vaccine candidate. Similarly, due to exposure to seasonal human coronaviruses, most humans also possess some degree of immunity against the COVID-19 virus. Therefore, pre-existing immunity to coronaviruses may also impact (positively or negatively) COVID-19 vaccines that are currently being tested in clinical trials.

## 4. Mechanisms Underlying Suppression of LAIV Immunogenicity by Pre-Existing Immunity: Alternative Vaccination Strategies to Circumvent this Issue

It is important to note that in children aged 6 months–17 years, LAIV demonstrated higher protective efficacy against culture-confirmed influenza illness compared to IIV [18,55,56]. Two doses of trivalent LAIV in a single year followed by a single dose of trivalent LAIV in a subsequent year elicited robust protective efficacy against circulating influenza strains [77,78]. More importantly, similar sequential vaccination with trivalent LAIV also afforded protective efficacy against a drifted circulating H3N2 strain in year two of vaccination in children [75,90]. The latter study also detected cross-reactive antibodies against H3N2 viruses following LAIV vaccination [75]. A retrospective study has shown that multiple vaccinations with LAIV over a period of three years afforded more than 50% protective efficacy against H1N1 as well as cross-protection against H3N2 viruses [107]. A possible explanation for the observed cross-protection in young children is that the first exposure to the influenza virus via LAIV establishes cross-reactive T cell immunity in influenza naïve individuals and subsequent LAIV vaccinations further increase or maintain T cell immunity at protective levels [74]. Thus, the first influenza vaccination in the form of LAIV may be a better vaccination approach compared to IIV first followed by LAIV. However, current influenza vaccination policy recommends IIV over LAIV for infants. This recommendation is based on early clinical trials that have found an association between LAIV and increased wheezing and asthma attacks in young children below the age of 2 years [108,109]. Thus, LAIV is indicated for use in individuals 2 to 49 years of age. However, considering the potential benefits of LAIV, perhaps LAIV should be offered as a first influenza vaccine for infants with no family history of wheezing and/or allergic asthma. Future research is necessary to determine if initial vaccination with LAIV, followed by IIV, establishes better mucosal T cell immunity. Alternatively, certain adjuvants could be a component of IIV to promote cellular immunity. Squalene-based adjuvants, such as MF59, have been approved for human use in the U.S. and have been shown to improve T-cell immunogenicity of IIV in humans [110]. Thus, in a sequential vaccination setting, adjuvanted IIV may circumvent the negative impact of pre-existing antibodies. 

Altered vaccine antigen trafficking by pre-existing antibodies may contribute to the reduced efficacy of LAIV. Preliminary studies in our laboratory have revealed that the presence of alveolar macrophages (AMs) is associated with reduced vaccine efficacy. In our mouse model, depletion of AMs at the time of vaccination significantly increased the efficacy of intranasal (i.n.) administered IIV (unpublished data). Likewise, it is possible that the reduced LAIV efficacy in IIV vaccinated mice may entail alveolar macrophages. AMs in a steady state are thought to play a critical role in maintaining lung homeostasis. AMs are effective in clearing inhaled antigens without initiating potentially deleterious immune responses. This is important given that AMs represent more than 95% of immune cells found in the airway and therefore they are the first to contact inhaled particles [111]. AMs can express cytokines such as transforming growth factor β1 (TGF-β1), prostaglandin E_2_ (PGE_2_), and platelet-activating factor (PAF), which are involved in immune suppression [112,113,114]. Thus, it is plausible that LAIV viruses opsonized by pre-existing antibodies are taken up by the AMs, as opposed to epithelial cells, which in turn activates immunosuppressive mechanisms. Studies have shown that TGF-β expression by AMs leads to activation of regulatory T cells (Tregs), which are important in mediating suppression of the immune response [115,116,117]. Delineating immune suppressive mechanisms may potentially facilitate alternative strategies to circumvent detrimental processes and mitigate the negative impact of pre-existing antibodies on LAIV efficacy. 

In addition to evaluating immune suppressive events, understanding the impact of pre-existing immunity on LAIV-infectivity of lung epithelia and antigen presentation by dendritic cells is important. Successful replication of LAIV in nasal epithelial cells induces pro-inflammatory type 1 (Th1) associated chemokines including chemokine (c-x-c motif) ligand (CXCL) 9 and CXCL11 in the lungs [85,118]. Such chemotactic mediators cause cellular infiltration by antigen presenting cells (APCs) such as CD11b^-^ dendritic cells (DCs), which are mainly known for their cross-presentation of antigens to T cells, favoring Th1 responses for influenza viral clearance [119]. Furthermore, studies using influenza infection models have shown differential roles of CD11b^+^ conventional DCs (cDCs) and lung resident and cross-presenting CD11b^-^ cDCs in the activation of cross-protective T responses in mice [120,121,122,123]. Thus, neutralization of LAIV by pre-existing antibodies potentially prevents successful infection of nasal epithelia causing reduced Th1 inflammation followed by suboptimal DC recruitment. In addition to pro-inflammatory innate immune responses in the lung mucosae, lung resident memory T and B cells have recently emerged as key players in memory responses against influenza. Recently discovered lung resident memory B cells (BRMs) and tissue resident memory T cells (TRMs) have been described to participate in cross-protection against influenza infection [4,7,124,125]. Therefore, investigating the impact of pre-existing immunity on the establishment of lung resident memory lymphocytes may also enrich our understanding of detrimental effects of pre-existing immunity on the cross-protective efficacy of LAIV. 

## 5. Summary and Conclusions

In conclusion, pre-existing immunity to influenza significantly influences LAIV efficacy against influenza viruses; therefore, contributing towards the consistently observed variable efficacies of LAIV in clinical trials. Pre-existing antibodies derived from IIV have been shown to inhibit influenza viral replication causing diminished memory CD8^+^ T cell responses, which are crucial for cross-protective immunity against antigenically drifted influenza strains. Hence, a similar mechanism may be contributing to the observed ablation of cross-protective efficacy of LAIV in our recent IIV prime/LAIV boost study in mice. However, the precise mechanism by which pre-existing antibodies suppress T cell immunogenicity of LAIV is not fully understood. Further advancement in this field is clearly needed to develop a better vaccination approach that can circumvent the negative impact of IIV on LAIV efficacy.

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
