# Peer review of "Impact of Pre-Existing Immunity to Influenza on Live-Attenuated Influenza Vaccine (LAIV) Immunogenicity"

_vaccines, 2020, doi:10.3390/vaccines8040683_

Round 1

Reviewer 1 Report

Very interesting article that discussed the impact of pre-existing immunity to influenza on LAIV immunogenicity. Although authors have discussed the alternative solutions but it would further improve the article if authors provide some discussion on subunit based vaccine for T cell immunity as it will not be impacted by pre-existing immunity.  

Author Response

Reviewer #1:

Very interesting article that discussed the impact of pre-existing immunity to influenza on LAIV immunogenicity. Although authors have discussed the alternative solutions but it would further improve the article if authors provide some discussion on subunit based vaccine for T cell immunity as it will not be impacted by pre-existing immunity. 

Authors’ response:

We appreciate the reviewer’s insightful comment. Information pertaining to this has now been added to the manuscript.

Lines 189-192: “Alternatively, certain adjuvants could be a component of IIV to promote cellular immunity. Squalene-based adjuvants, such as MF59, have been approved for human use in the U.S. and have been shown to improve T-cell immunogenicity of IIV in humans [110]. Thus, in a sequential vaccination setting, adjuvanted IIV may circumvent the negative impact of pre-existing antibodies.”

Reviewer 2 Report

Protection against influenza has become all the more important in the face of the SARS-CoV-2 pandemic, and pre-existing immunity against SARS-CoV-2 has become an "hot topic". Therefore, this timely review of pre-existing immunity against influenza has some added value.

The field of influenze immunity and immunisation is a complex one, and the authors have successfully brought together citations which are both balanced and informative of clinical decision-making. The exploration of immunity in the face of previous influenza vaccination is of particular interest.

It will be of particular interest to re-explore the phenomena and mechanisms described in the review, once the SARS-CoV-2 pandemic is under control, especially if viral interference has been observed and/or social distancing, the wearing of masks and lockdowns have reduced the overall population exposure to influenza.

Whilst it is beyond the scope of the review, the authors may like to go through the manuscrip to see if there are any messages/warnings which may pertain to the multitude of SARS-CoV2 vaccines promised for 2021.

Author Response

Reviewer #2:

Protection against influenza has become all the more important in the face of the SARS-CoV-2 pandemic, and pre-existing immunity against SARS-CoV-2 has become an "hot topic". Therefore, this timely review of pre-existing immunity against influenza has some added value.

The field of influenza immunity and immunisation is a complex one, and the authors have successfully brought together citations which are both balanced and informative of clinical decision-making. The exploration of immunity in the face of previous influenza vaccination is of particular interest.

It will be of particular interest to re-explore the phenomena and mechanisms described in the review, once the SARS-CoV-2 pandemic is under control, especially if viral interference has been observed and/or social distancing, the wearing of masks and lockdowns have reduced the overall population exposure to influenza.

Whilst it is beyond the scope of the review, the authors may like to go through the manuscrip to see if there are any messages/warnings which may pertain to the multitude of SARS-CoV2 vaccines promised for 2021.

Authors’ response: We agree with the reviewer’s advice and have now added following sentences:

Line 160-166: “This is particularly true now given the ongoing COVID-19 pandemic. One of the COVID-19 vaccines under development in Hong Kong uses live attenuated influenza virus expressing SARS-CoV-2 antigens [106]. We speculate that anti-vector immunity to influenza may also be detrimental to the immunogenicity of this vaccine candidate. Similarly, due to exposures to seasonal human coronaviruses, most humans also possess some degree of immunity against the COVID-19 virus.  Therefore, pre-existing immunity to coronaviruses may also impact (positively or negatively) COVID-19 vaccines that are currently being tested in clinical trials.”

Reviewer 3 Report

This work builds from the authors ongoing research efforts with a most recent publication highlighting their thesis that pre-existing antibodies alter immunogenicity of LAIV. (Vaccines (Basel). 2020 Sep; 8(3): 459.)

This commentary is well written and interesting with a logical progression of discussion from protective immune correlates to LAIV efficacy, to the role of pre-existing immunity with alternative vaccine strategies proposed. 

One minor edit:
Line 170: "...may be a better..."

Author Response

Reviewer #3:

This work builds from the authors ongoing research efforts with a most recent publication highlighting their thesis that pre-existing antibodies alter immunogenicity of LAIV. (Vaccines (Basel). 2020 Sep; 8(3): 459.) This commentary is well written and interesting with a logical progression of discussion from protective immune correlates to LAIV efficacy, to the role of pre-existing immunity with alternative vaccine strategies proposed. 

One minor edit: Line 170: "...may be a better..."

Authors’ response: Line 181 has now been corrected and highlighted in yellow in the manuscript.

Reviewer 4 Report

Greater Importance: 

  1. Comments in manuscript about broader repertoire of antigens in LAIV compared to IIV I think must be caveated with the comment that the background will be on a different backbone than the circulating virus, although I agree that there will be more antigens available for immune exposure than IIV. 
  2. "non-HA inhibiting (HI) antibodies". I think this comment would make more sense to say that it has been shown that in one strain of influenza, H7N9, that LAIV has also been able to induce immunity vis ADCC and CDL. 
  3. Comment on pre-existing immunity as potential factor to influencing LAIV efficacy. As this is the primary purpose of the paper, I think it makes sense to present a hypothesis (negative interference is main driver of decreased efficacy) and the other possible alternatives (fitness issues with cold adapted backbone in humans vs animal models, vaccine production issues, vaccine mismatch, etc).
  4. Would not comment that LAIV is "highly efficacious" as this does not have a definition and is misleading. I would either say that it is superior to IIV or comment on a specific efficacy estimation that has been published. 

Minor Points: 

  1. Comments throughout the manuscript that discuss cross reactivity should be detailed if it was T cell mediated cross reactivity or humoral cross reactivity. 
  2. Style comment in reference to Cytotoxic CD4 T cells. I think that the concept presented is a good one but it might flow better if presented as CD4 helper T, Cd8 cytotoxic and then present the newer concept of CD4 cytotoxic and how that may be involved in the immune response. 
  3. "A possible explanation for observed cross protection in the young children is that first exposure to influenza virus via LAIV establishes cross reactive T cell immunity. I think this is a great theory although I think it might be nice to include that these studies have not been done and likely will not be achieved without a clinical trial setting due to current CDC guidelines for LAIV. 

Author Response

Reviewer #4:

Greater Importance: 

Comment 1: Comments in manuscript about broader repertoire of antigens in LAIV compared to IIV I think must be caveated with the comment that the background will be on a different backbone than the circulating virus, although I agree that there will be more antigens available for immune exposure than IIV. 

R4 Answer 1: We apologize for the confusion. We clarified in the text as follows:

Line 36-40: Since LAIV is a whole virus vaccine, it presents a broader repertoire of antigens, including internal proteins that are highly conserved among influenza A viruses. In contrast, IIV consists of primarily two viral surface antigens: HA and NA, which are highly variable. Consequently, LAIV can elicit a broader immune response and theoretically, superior cross-protective immunity compared to IIV [reviewed in [13]].

Comment 2: "non-HA inhibiting (HI) antibodies". I think this comment would make more sense to say that it has been shown that in one strain of influenza, H7N9, that LAIV has also been able to induce immunity vis ADCC and CDL. 

Authors’ response: We have taken the reviewer’s advice and changed lines 62-64 to: “Furthermore, LAIV has been shown to elicit non-neutralizing antibodies against H7N9 strain, which may contribute to cross-protection potentially via antibody-dependent cellular cytotoxicity (ADCC) mediated by Fc-FcR interactions”.

Comment 3: Comment on pre-existing immunity as potential factor to influencing LAIV efficacy. As this is the primary purpose of the paper, I think it makes sense to present a hypothesis (negative interference is main driver of decreased efficacy) and the other possible alternatives (fitness issues with cold adapted backbone in humans vs animal models, vaccine production issues, vaccine mismatch, etc).

Authors’ response: As per the reviewer’s advice we have now included our central hypothesis in lines 129-131: “Thus, taken together we hypothesize that pre-existing antibodies derived from IIV vaccination and/or seasonal influenza infections neutralizes LAIV viruses and prevents establishment of mucosal immunity at the vaccination site.

Comment 4: Would not comment that LAIV is "highly efficacious" as this does not have a definition and is misleading. I would either say that it is superior to IIV or comment on a specific efficacy estimation that has been published. 

R4 Answer 4: The reviewer raised an important point. We have now added this information in lines 169-170: “LAIV demonstrated higher protective efficacy against culture-confirmed influenza illness compared to IIV”.

Minor Points: 

Comment 5: Comments throughout the manuscript that discuss cross reactivity should be detailed if it was T cell mediated cross reactivity or humoral cross reactivity. 

Authors’ response: According to the reviewer’s comment, wherever applicable, information about mechanisms mediating observed cross-protective efficacy have been added in lines 42, 44, 66-67, and 174-175.

Comment 6: Style comment in reference to Cytotoxic CD4 T cells. I think that the concept presented is a good one but it might flow better if presented as CD4 helper T, Cd8 cytotoxic and then present the newer concept of CD4 cytotoxic and how that may be involved in the immune response. 

Authors’ response: As per reviewer’s comment, in lines 49-55, the order of appearance of helper CD4 T cells, cytotoxic CD8 T cells and cytotoxic CD4 T cells in the paragraph have now been altered.

Comment 7: A possible explanation for observed cross protection in the young children is that first exposure to influenza virus via LAIV establishes cross reactive T cell immunity. I think this is a great theory although I think it might be nice to include that these studies have not been done and likely will not be achieved without a clinical trial setting due to current CDC guidelines for LAIV. 

Authors’ response: Per request, the following was added to the text:

Line 187-189: “Future research is necessary to determine if initial vaccination with LAIV, followed by IIV, establishes better mucosal T cell immunity.”